# Is a picture worth a thousand words?
# A Deep Multi-Modal Fusion Architecture for
# Product Classification in e-commerce

**Tom Zahavy & Shie Mannor**
Department of Electrical Engineering
The Technion - Israel Institute of Technology
Haifa 32000, Israel
{tomzahavy@tx,shie@ee}.technion.ac.il

**Alessandro Magnani & Abhinandan Krishnan**

Walmart Labs
Sunnyvale, California
{AMagnani,AKrishnan}@walmartlabs.com

## Abstract

Classifying products into categories precisely and efficiently is a major challenge in modern e-commerce. The high traffic of new products uploaded daily and the dynamic nature of the categories raise the need for machine learning models that can reduce the cost and time of human editors. In this paper, we propose a decision level fusion approach for multi-modal product classification using text and image inputs. We train input specific state-of-the-art deep neural networks for each input source, show the potential of forging them together into a multi-modal architecture and train a novel policy network that learns to choose between them. Finally, we demonstrate that our multi-modal network improves the top-1 accuracy % over both networks on a real-world large-scale product classification dataset that we collected from Walmart.com. While we focus on image-text fusion that characterizes e-commerce domains, our algorithms can be easily applied to other modalities such as audio, video, physical sensors, etc.

## 1 Introduction

Product classification is a key issue in e-commerce domains. A product is typically represented by metadata such as its title, image, color, weight and so on, and most of them are assigned manually by the seller. Once a product is uploaded to an e-commerce website, it is typically placed in multiple categories. Categorizing products helps e-commerce websites to provide costumers a better shopping experience, for example by efficiently searching the products catalog or by developing recommendation systems. A few examples of categories are internal taxonomies (for business needs), public taxonomies (such as groceries and office equipment) and the product's shelf (a group of products that are presented together on an e-commerce web page). These categories vary with time in order to optimize search efficiency and to account for special events such as holidays and sports events. In order to address these needs, e-commerce websites typically hire editors and use crowdsourcing platforms to classify products. However, due to the high amount of new products uploaded daily and the dynamic nature of the categories, machine learning solutions for product classification are very appealing as means to reduce the time and economic costs. Thus, precisely categorizing items emerges as a significant issue in e-commerce domains.

A shelf is a group of products presented together on an e-commerce website page, and usually contain products with a given theme/category (e.g., Women boots, folding tables). Product to shelf classification is a challenging problem due to data size, category skewness, and noisy metadata and labels. In particular, it presents three fundamental challenges for machine learning algorithms. First, it is typically a multi-class problem with thousands of classes. Second, a product may belong to multiple shelves making it a multi-label problem. And last, a product has both an image and a text input making it a multi-modal problem.

Products classification is typically addressed as a text classification problem because most metadata of items are represented as textual features (Pyo et al., 2010). Text classification is a classic topic for natural language processing, in which one needs to assign predefined categories to text inputs.

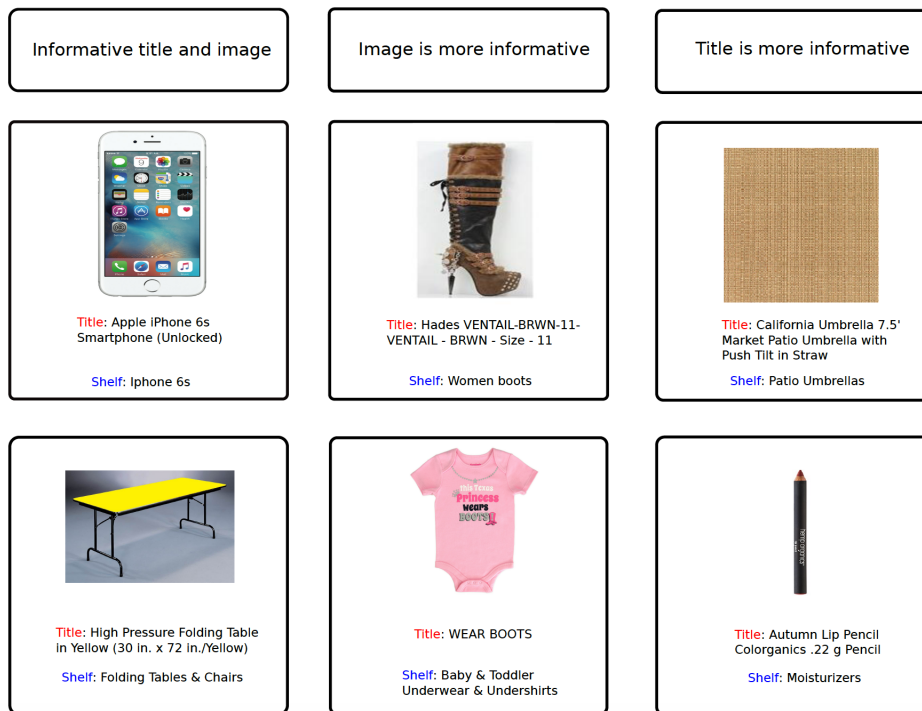

Figure 1: Predicting shelves from product metadata obtained from Walmart.com. **Left:** products that have both an image and a title that contain useful information for predicting the product's shelf. **Center, top:** the boots title gives specific information about the boots but does not mention that the product is a boot, making it harder to predict the shelf. **Center, bottom:** the baby toddler shirt's title is only refers to the text on the toddler shirt and does not mention that it is a product for babies. **Right, top:** the umbrella image contains information about its color but it is hard to understand that the image is referring to an umbrella. **Right, bottom:** the lips pencil image looks like a regular pencil, making it hard to predict that it belongs to the moisturizers shelf.

Standard methods follow a classical two-stage scheme of extraction of (handcrafted) features, followed by a classification stage. Typical features include bag-of-words or n-grams, and their TF-IDF. On the other hand, Deep Neural Networks use generic priors instead of specific domain knowledge (Bengio et al., 2013) and have been shown to give competitive results on text classification tasks (Zhang et al., 2015). In particular, Convolutional neural networks (CNNs) (Kim, 2014; Zhang et al., 2015; Conneau et al., 2016) and Recurrent NNs (Lai et al., 2015; Pyo et al., 2010; Xiao & Cho, 2016) can efficiently capture the sequentiality of the text. These methods are typically applied directly to distributed embedding of words (Kim, 2014; Lai et al., 2015; Pyo et al., 2010) or characters (Zhang et al., 2015; Conneau et al., 2016; Xiao & Cho, 2016), without any knowledge on the syntactic or semantic structures of a language. However, all of these architectures were only applied on problems with a small amount of labels ($\sim 20$) while e-commerce shelf classification problems typically have thousands of labels with multiple labels per product.

In Image classification, CNNs are widely considered the best models, and achieve state-of-the-art results on the ImageNet Large-Scale Visual Recognition Challenge (Russakovsky et al., 2015; Krizhevsky et al., 2012; Simonyan & Zisserman, 2014; He et al., 2015). However, as good as they are, the classification accuracy of machine learning systems is often limited in problems with many classes of object categories. One remedy is to leverage data from other sources, such as text data. However, the studies on multi-modal deep learning for large-scale item categorization are still rare to the best of our belief. In particular in a setting where there is a significant difference in discriminative power between the two types of signals.

In this work, we propose a multi-modal deep neural network model for product classification. Our design principle is to leverage the specific prior for each data type by using the current state-of-

the-art classifiers from the image and text domains. The final architecture has 3 main components (Figure 2, Right): a text CNN (Kim, 2014), an image CNN (Simonyan & Zisserman, 2014) and a policy network that learns to choose between them. We collected a large-scale data set of 1.2 million products from the Walmart.com website. Each product has a title and an image and needs to be classified to a shelf (label) with 2890 possible shelves. Examples from this dataset can be seen in Figure 1 and are also available on-line at the Walmart.com website. For most of the products, both the image and the title of each product contain relevant information for customers. However, it is interesting to observe that for some of the products, both input types may not be informative for shelf prediction (Figure 1). This observation motivates our work and raises interesting questions: which input type is more useful for product classification? is it possible to forge the inputs into a better architecture?

In our experiments, we show that the text CNN outperforms the image one. However, for a relatively large number of products ($\sim 8\%$), the image CNN is correct while the text CNN is wrong, indicating a potential gain from using a multi-modal architecture. We also show that the policy is able to choose between the two models and give a performance improvement over both state-of-the-art networks.

To the best of our knowledge, this is the first work that demonstrates a performance improvement on top-1 classification accuracy by using images and text on a large-scale classification problem. In particular, our main contributions are:

- We demonstrate that the text classification CNN (Kim, 2014) outperforms the VGG network (Simonyan & Zisserman, 2014) on a real-world large-scale product to shelf classification problem.

- We analyze the errors made by the different networks and show the potential gain of multi-modality.

- We propose a novel decision-level fusion policy that learns to choose between the text and image networks and improve over both.

## 2 MULTI-MODALITY

Over the years, a large body of research has been devoted to improving classification using ensembles of classifiers (Kittler et al., 1998; Hansen & Salamon, 1990). Inspired by their success, these methods have also been used in multi-modal settings (e.g.,Guillaumin et al. (2010); Poria et al. (2016)), where the source of the signals, or alternatively their modalities, are different. Some examples include audio-visual speech classification (Ngiam et al., 2011), image and text retrieval (Kiros et al.), sentiment analysis and semi-supervised learning (Guillaumin et al., 2010).

Combining classifiers from different input sources presents multiple challenges. First, classifiers vary in their discriminative power, thus, an optimal unification method should be able to adapt itself for specific combinations of classifiers. Second, different data sources have different state-of-the-art architectures, typically deep neural networks, which vary in depth, width, and optimization algorithm; making it non-trivial to merge them. Moreover, a multi-modal architecture potentially has more local minima that may give unsatisfying results. Finally, most of the publicly available real-world big data classification datasets, an essential building block of deep learning systems, typically contain only one data type.

Nevertheless, the potential performance boost of multi-modal architectures has motivated researchers over the years. Frome et al. (2013) combined an image network (Krizhevsky et al., 2012) with a Skip-gram Language Model in order to improve classification results on ImageNet. However, they were not able to improve the top-1 accuracy prediction, possibly because the text input they used (image labels) didn't contain a lot of information. Other works, used multi-modality to learn good embedding but did not present results on classification benchmarks (Lynch et al., 2015; Kiros et al.; Gong et al., 2014). Kannan et al. (2011) suggested to improve text-based product classification by adding an image signal, training an image classifier and learning a decision rule between the two. However, they only experimented with a small dataset and a low number of labels, and it is not clear how to scale their method for extreme multi-class multi-label applications that characterize real-world problems in e-commerce.

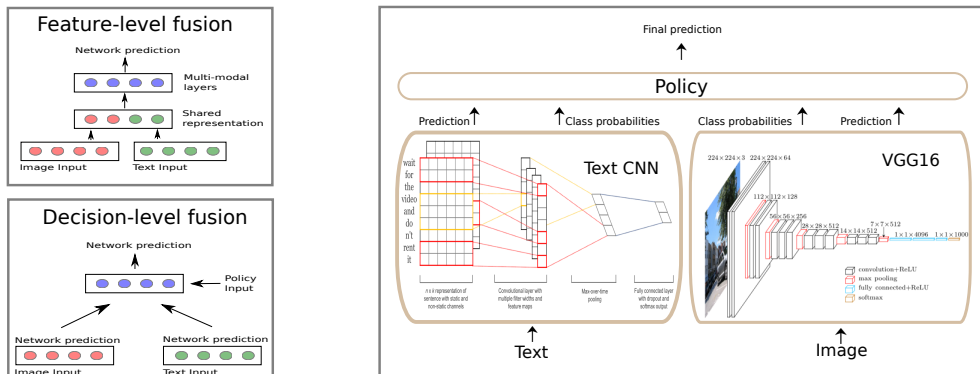

Figure 2: **Multi-modal fusion architectures.**
**Left, top: Feature-level fusion.** Each modality is processed in a different pipe. After a certain depth, the pipes are concatenated followed by multi-modal layers. **Left, bottom: Decision-level fusion.** Each modality is processed in a different pipe and gives a prediction. A policy network is learning to decide which classifier to use. **Right: The proposed multi-modal architecture.**

Adding modalities can improve the classification of products that have a non-informative input source (e.g., image or text). In e-commerce, for example, classifiers that rely exclusively on text suffer from short and non-informative titles, differences in style between vendors and overlapping text across categories (i.e., a word that helps to classify a certain class may appear in other classes). Figure 1 presents a few examples of products that have only one informative input type. These examples suggest that a multi-modal architecture can potentially outperform a classifier with a single input type.

Most unification techniques for multi-modal learning are partitioned between feature-level fusion techniques and decision-level fusion techniques (Figure 2, Left).

## 2.1 FEATURE LEVEL FUSION

Feature-level fusion is characterized by three phases: (a) learning a representation, (b) supervised training, and (c) testing. The different unification techniques are distinguished by the availability of the data in each phase (Guillaumin et al., 2010). For example, in cross-modality training, the representation is learned from all the modalities, but only one modality is available for supervised training and testing. In other cases, all of the modalities are available at all stages but we may want (or not) to limit their usage given a certain budget. Another source for the distinction is the order in which phases (a) and (b) are made. For example, one may first learn the representation and then learn a classifier from it, or learn both the representation and the classifier in parallel. In the deep learning context, there are two common approaches. In the first approach, we learn an end-to-end deep NN; the NN has multiple input-specific pipes that include a data source followed by input specific layers. After a certain depth, the pipes are concatenated followed by additional layers such that the NN is trained end-to-end. In the second approach, input specific deep NNs are learned first, and a multi-modal representation vector is created by concatenating the input specific feature vectors (e.g., the neural network's last hidden layer). Then, an additional classifier learns to classify from the multi-modal representation vector. While multi-modal methods have shown potential to boost performance on small datasets (Poria et al., 2016), or on top-k accuracy measures (Frome et al., 2013), we are not familiar with works that succeeded with applying it on a large-scale classification problem and received performance improvement in top-1 accuracy.

## 2.2 DECISION-LEVEL FUSION

In this approach, an input specific classifier is learned for each modality, and the goal is to find a decision rule between them. The decision rule is typically a pre-defined rule (Guillaumin et al., 2010) and is not learned from the data. For example, Poria et al. (2016) chose the classifier with the maximal confidence, while Krizhevsky et al. (2012) average classifier predictions. However, in this work we show that learning the decision rule yields significantly better results on our data.

## 3 METHODS AND ARCHITECTURES

In this section, we give the details of our multi-modal product classification architecture. The architecture is composed of a text CNN and an image CNN which are forged together by a policy network, as can be seen in Figure 2, Right.

### 3.1 MULTI LABEL COST FUNCTION

Our cost function is the weighted sigmoid cross entropy with logits, a common cost function for multi-label problems. Let $x$ be the logits, $z$ be the targets, $q$ be a positive weight coefficient, used as a multiplier for the positive targets, and $\sigma(x) = \frac{1}{1+exp(-x)}$. The loss is given by:

$$\text{Cost(x,z;q)} = -qz \cdot \log(\sigma(x)) - (1-z) \cdot \log(1 - \sigma(x)) =$$
$$(1-z) \cdot x + (1 + (q-1) \cdot z) \cdot \log(1 + exp(-x)).$$

The positive coefficient $q$, allows one to trade off recall and precision by up- or down-weighting the cost of a positive error relative to a negative error. We found it to have a significant effect in practice.

### 3.2 TEXT CLASSIFICATION

For the text signal, we use the text CNN architecture of Kim (2014). The first layer embeds words into low-dimensional vectors using random embedding (different than the original paper). The next layer performs convolutions over time on the embedded word vectors using multiple filter sizes (3, 4 and 5), where we use $128$ filters from each size. Next, we max-pool-over-time the result of each convolution filter and concatenated all the results together. We add a dropout regularization layer (0.5 dropping rate), followed by a fully connected layer, and classify the result using a softmax layer. An illustration of the Text CNN can be seen in Figure 2.

### 3.3 IMAGE CLASSIFICATION

For the image signal, we use the VGG Network (Simonyan & Zisserman, 2014). The input to the network is a fixed-size 224 x 224 RGB image. The image is passed through a stack of convolutional layers with a very small receptive field: 3 x 3. The convolution stride is fixed to 1 pixel; the spatial padding of the convolutional layer is 1 pixel. Spatial pooling is carried out by five max-pooling layers, which follow some of the convolutional layers. Max-pooling is performed over a 2 x 2 pixel window, with stride 2. A stack of convolutional layers is followed by three Fully-Connected (FC) layers: the first two have 4096 channels each, the third performs 2890-way product classification and thus contains 2890 channels (one for each class). All hidden layers are followed by a ReLu non-linearity. The exact details can be seen in Figure 2.

### 3.4 MULTI-MODAL ARCHITECTURE

We experimented with four types of multi-modal architectures. *(1)* Learning decision-level fusion policies from different inputs. *(1a)* Policies that use the text and image CNNs **class probabilities** as input (Figure 2). We experimented with architectures that have one or two fully connected layers (the two-layered policy is using 10 hidden units and a ReLu non-linearity between them). *(1b)* Policies that use the **text and/or image** as input. For these policies, the architecture of policy network was either the text CNN or the VGG network. In order to train policies, labels are collected from the image and text networks predictions, i.e., the label is 1 if the image network made a correct prediction while the text network made a mistake, and 0 otherwise. On evaluation, we use the policy predictions to select between the models, i.e., if the policy prediction is 1 we use the image network, and use the text network otherwise. *(2)* Pre-defined policies that average the predictions of the different CNNs or choose the CNN with the highest confidence. *(3)* End-to-end feature-level fusion, each input type is processed by its specific CNN. We concatenate the last hidden layers of the CNNs and add one or two fully connected layers. All the layers are trained together end-to-end (we also tried to initialize the input specific weights from pre-trained single-modal networks). *(4)* Multi-step feature-level fusion. As in (3), we create shared representation vector by concatenating the last hidden layers. However, we now keep the shared representation fixed and learn a new classifier from it.

## 4 EXPERIMENTS

### 4.1 SETUP

Our dataset contains 1.2 million products (title image and shelf) that we collected from Walmart.com (offered online and can be viewed at the website) and were deemed the hardest to classify by the current production system. We divide the data into training (1.1 million) validation (50k) and test (50k). We train both the image network and the text network on the training dataset and evaluate them on the test dataset. The policy is trained on the validation dataset and is also evaluated on the test dataset. The objective is to classify the product's shelf, from 2890 possible choices. Each product is typically assigned to more than one shelf (3 on average), and the network is considered accurate if its most probable shelf is one of them.

### 4.2 TRAINING THE TEXT ARCHITECTURE

**Preprocess:** we build a dictionary of all the words in the training data and embed each word using a random embedding into a one hundred dimensional vector. We trim titles with more than 40 words and pad shorter titles with nulls.

We experimented with different batch sizes, dropout rates, and filters stride, but found that the vanilla architecture (Kim, 2014) works well on our data. This is consistent with Zhang & Wallace (2015), who showed that text CNNs are not very sensitive to hyperparameters. We tuned the cost function positive coefficient parameter $q$, and found out that the value 30 performed best in practice (we will also use this value for the image network). The best CNN that we trained classified $70.1\%$ of the products from the test set correctly (Table 1).

### 4.3 TRAINING THE IMAGE ARCHITECTURE

**Preprocess:** we re-size all the images into 224 x 224 pixels and reduce the image mean.

The VGG network that we trained classified $57\%$ of the products from the test set correctly. This is a bit disappointing if we compare it to the performance of the VGG network on ImageNet ($\sim 75\%$). There are a few differences between these two datasets that may explain this gap. First, our data has 3 times more classes and contains multiple labels per image making the classification harder, and second, Figure 1 implies that some of our images are not informative for shelf classification. Some works claim that the features learned by VGG on ImageNet are global feature extractors (Lynch et al., 2015). We therefore decided to use the weights learned by VGG on ImageNet and learn only the last layer. This configuration yielded only $36.7\%$ accuracy. We believe that the reason is that some of the ImageNet classes are irrelevant for e-commerce (e.g., vehicles and animals) while some relevant categories are misrepresented (e.g., electronics and office equipment). It could also be that our images follow some specific pattern of white background, well-lit studio etc., that characterizes e-commerce.

### 4.4 ERROR ANALYSIS

Is a picture worth a thousand words? Inspecting Figure 3, we can see that the text network outperformed the image network on this dataset, classifying more products correctly. Similar results were reported before (Pyo et al., 2010; Kannan et al., 2011) but to the best of our knowledge, this is the first work that compares state-of-the-art text and image CNNs on a real-world large-scale e-commerce dataset.

What is the potential of multi-modality? We identified that for $7.8\%$ of the products the image network made a correct prediction while the text network was wrong. This observation is encouraging since it implies that there is a relative big potential to harness via multi-modality. We find this large gap surprising since different neural networks applied to the same problem tend to make the same mistakes (Szegedy et al., 2013).

Unification techniques for multi-modal problems typically use the last hidden layer of each network as features (Frome et al., 2013; Lynch et al., 2015; Pyo et al., 2010). We therefore decided to visualize the activations of this layer using a tSNE map (Maaten & Hinton, 2008). Figure 3, depicts such a map for the activations of the text model (the image model yielded similar results). In particular,

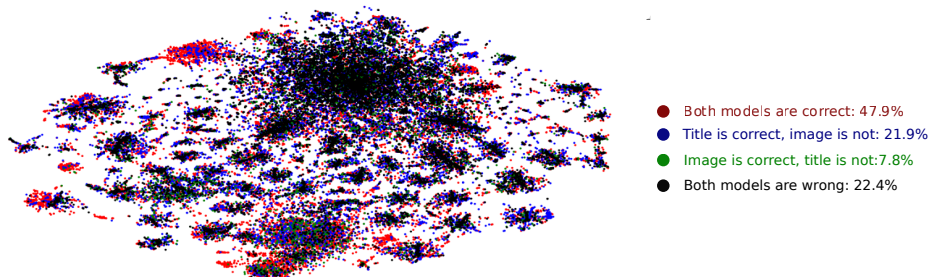

Figure 3: Error analysis using a tSNE map, created from the last hidden layer neural activations of the text model.

we were looking for regions in the tSNE map where the image predictions are correct and the text is wrong (Figure 3, green). Finding such a region will imply that a policy network can learn good decision boundaries. However, we can see that there are no well-defined regions in the tSNE maps where the image network is correct and the title is wrong (green), thus implying that it might be hard to identify these products using the activations of the last layers.

## 4.5 Multi-modal Unification Techniques

Our error analysis experiment highlights the potential of merging image and text. Still, we found it hard to achieve the upper bound provided by the error analysis in practice. We now describe the policies that managed to achieve performance boost in top-1 accuracy % over the text and image networks, and then provide discussion on other approaches that we tried but didn't work.

**Decision-level fusion:** We trained policies from different data sources (e.g., title, image, and each CNN class probabilities), using different architectures and different hyperparameters. Looking at Table 1, we can see that the best policies were trained using the class probabilities (the softmax probabilities) of the image and text CNNs as inputs. The amount of class probabilities that were used (top-1, top-3 or all) did not have a significant effect on the results, indicating that the top-1 probability contains enough information to learn good policies. This result makes sense since the top-1 probability measures the confidence of the network in making a prediction. Still, the top-3 probabilities performed slightly better, indicating that the difference between the top probabilities may also matter. We can also see that the 2-layer architecture outperformed the 1-layer, indicating that a linear policy is too simple, and deeper models can yield better results. Last, the cost function positive coefficient q had a big impact on the results. We can see that for $q = 1$, the policy network is more accurate in its prediction however it achieves worse results on shelf classification. For $q = 5$ we get the best results, while higher values of $q$ (e.g., 7 or 10) resulted in inaccurate policies that did not perform well in practice.

| Policy input | # layers | q | Text | Image | Policy | Oracle | Policy accuracy |
|---|---|---|---|---|---|---|---|
| CP-1 | 1 | 5 | 70.1 | 56.7 | 71.4 (+1.3) | 77.5 (+7.8) | 86.4 |
| CP-1 | 2 | 5 | 70.1 | 56.6 | 71.5 (+1.4) | 77.6 (+7.5) | 84.2 |
| CP-all | 2 | 5 | 70.1 | 56.6 | 71.4 (+1.3) | 77.6 (+7.5) | 84.6 |
| **CP-3** | **2** | **5** | **70.2** | **56.7** | **71.8 (+1.6)** | **77.7 (+7.5)** | **84.2** |
| CP-3 | 2 | 1 | 70.2 | 56.7 | 70.2 (+0) | 77.7 (+7.5) | 92.5 |
| CP-3 | 2 | 7 | 70.0 | 56.6 | 71.0 (+1.0) | 77.5 (+7.5) | 79.1 |
| CP-3 | 2 | 10 | 70.1 | 56.6 | 70.7 (+0.6) | 77.6 (+7.5) | 75.0 |
| Image | - | 5 | 70.1 | 56.6 | 68.5(-1.6) | 77.6 (+7.5) | 80.3 |
| Text | - | 5 | 70.1 | 56.6 | 69.0 (-1.1) | 77.6 (+7.5) | 83.7 |
| Both | - | 5 | 70.1 | 56.6 | 66.1 (-4) | 77.6 (+7.5) | 73.7 |
| Fixed-Mean | - | - | 70.1 | 56.7 | 65.4 (+0) | 77.6 (+7.5) | - |
| Fixed-Max | - | - | 70.1 | 56.7 | 60.1 (-10) | 77.7 (+7.6) | 38.2 |

Table 1: **Decision-level fusion results.** Each row presents a different policy configuration (defined by the policy input, the number of layers and the value of $q$), followed by the accuracy % of the image, text, policy and oracle (optimal policy) classifiers on the test dataset. The policy accuracy column presents the accuracy % of the policy in making correct predictions, i.e., choosing the image network when it made a correct prediction while the text network didn't. Numbers in $(+\cdot)$ refer to the performance gain over the text CNN. Class Probabilities (CP) refer to the number of class probabilities used as input.

While it may not seem surprising that combining text and image will improve accuracy, in practice we found it extremely hard to leverage this potential. To the best of our knowledge, this is the first work that demonstrates a direct performance improvement on top-1 classification accuracy from using images and text on a large-scale classification problem.

We experimented with pre-defined policies that do not learn from the data. Specifically, we tried to average the logits, following (Krizhevsky et al., 2012; Simonyan & Zisserman, 2014), and to choose the network with the maximal confidence following (Poria et al., 2016). Both of these experiments yielded significantly worse results, probably, since the text network is much more accurate than the image one (Table 1). We also tried to learn policies from the text and/or the image input, using a policy network which is either a text CNN, a VGG network or a combination. However, all of these experiments resulted in policies that overfit the data and performed worse than the title model on the test data (Table 1). We also experimented with early stopping criteria, various regularization methods (dropout, l1, l2) and reduced model size but none could make the policy network generalize.

**Feature-level fusion:** Training a CNN end-to-end can be tricky. For example, each input source has its own specific architecture, with specific learning rate and optimization algorithm. We experimented with training the network end-to-end, but also with first training each part separately and then learning the concatenated parts. We tried different unification approaches such as gating functions (Srivastava et al., 2015), cross products and a different number of fully connected layers after the concatenation. These experiments resulted in models that were inferior to the text model. While this may seem surprising, the only successful feature level fusion that we are aware of (Frome et al., 2013), was not able to gain accuracy improvement on top-1 accuracy.

## 5 CONCLUSIONS

In this work, we investigated a multi-modal multi-class multi-label product classification problem and presented results on a challenging real-world dataset that we collected from Walmart.com. We discovered that the text network outperforms the image network on our dataset, and observed a big potential of fusing text and image inputs. Finally, we suggested a multi-modal decision-level fusion approach that leverages state-of-the-art results from image and text classification and forges them into a multi-modal architecture that outperforms both.

State-of-the-art image CNNs are much larger than text CNNs, and take more time to train and to run. Thus, extracting image features during run time, or getting the image network predictions may be prohibitively expensive. In this context, an interesting observation is that feature level fusion methods require using the image signal for each product, while decision level fusion methods require using the image network selectively making them more appealing. Moreover, our experiments suggest that decision-level fusion performs better than feature-level fusion in practice.

Finally, we were only able to realize a fraction of the potential of multi-modality. In the future, we plan to investigate deeper policy networks and more sophisticated measures of confidence. We also plan to investigate ensembles of image networks (Krizhevsky et al., 2012) and text networks (Pyo et al., 2010). We believe that the insights from training policy networks will eventually lead us to train end to end differential multi-modal networks.

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
