# Peer review of "Is a picture worth a thousand words? A Deep Multi-Modal Fusion Architecture for Product Classification in e-commerce"

_ICLR 2017 — rejected_

[Public Comment · Jung-Woo Ha · 06 Dec 2016]
**please fix a cited paper**

Hi, I am Jung-Woo Ha and the authour of a paper you cited in your work. 

In the references of your manuscript, I think that "Hyuna Pyo, Jung-Woo Ha, and Jeonghee Kim. Large-scale item categorization in e-commerce using
multiple recurrent neural networks. 2010." should be changed into "Jung-Woo Ha, Hyuna Pyo, and Jeonghee Kim. Large-scale item categorization in e-commerce using
multiple recurrent neural networks. In Proceedings of KDD 2016."
The url is

[Official Review · AnonReviewer1 · rating 4 · confidence 4 · 16 Dec 2016]
**No Title**

This paper presents a system approach to combine multiple modalities to perform classification in a practical scenario (e-commerce).

In general, I find the proposed approach in the paper sound and solid, but do not see novelty in the paper: feature fusion and decision time fusion are both standard practices in multi-modal analysis, and the rest of the paper offers no surprise in implementing such approaches. This seems to be a better fit for venues that focus more on production systems, and seems to be a bad fit for ICLR where the focus is more on research of novel algorithms and theories.

[Official Review · AnonReviewer3 · rating 5 · confidence 4 · 19 Dec 2016]
**Practical large-scale multi-model architecture but lack technical novelty**

This paper introduces a large-scale multi-model product classification system. The model consists of three modules, Image CNN (VGG 16 architecture), text CNN (Kim 2014) and decision-level fusion policies. The authors have tried several fusion methods: including policies taking inputs from text and image CNN probabilities; choose either CNN; average the predictions; end-to-end training. Experimental results show that text CNN alone works better than image CNN and multi-model fusion can improve the accuracy by a small margin. It is a little bit surprising that end-to-end feature level fusion works worse than text CNN alone. The writing is clear and there are a lot of useful practical experiences of learning large-scale model. However, I lean toward rejecting the paper because the following:

1) No other dataset reported. The authors haven't mentioned releasing the walmart dataset and it is going to be really hard to reproduce the results without the dataset. 
2) Technical novelty is limited. All the decision-level fusion policies have been investigated by some previous methods before. 
3) Performance gain is also limited.

[Official Review · AnonReviewer2 · rating 5 · confidence 4 · 19 Dec 2016]

This paper tackles the problem of multi-modal classification of text and images.

Pros:
- Interesting dataset and application.

Cons:
- The results are rather lacklustre, showing a very mild improvement compared to the oracle improvement. But perhaps some insights as to whether the incorrect decisions are humanly possible would help with significance of the results.
- Could have explored some intermediate architectures such as feature fusion + class probabilities with/without finetuning. There are no feature fusion results reported.
- No evaluation on standard datasets or comparison to previous works.

What is the policy learnt for CP-1? Given 2 input class probabilities, how does the network perform better than max or mean?

[Final Decision · Program Chairs · 06 Feb 2017]
**ICLR committee final decision**

Three knowledgable reviewers recommend rejection. While the application is interesting and of commercial value, the technical contribution falls below the ICLR's bar. I encourage the authors to improve the paper and submit it to a future conference.